# Hepatitis C Core Protein Induces a Genotype-Specific Susceptibility of Hepatocytes to TNF-Induced Death In Vitro and In Vivo

**DOI:** 10.3390/v14112521

**Published:** 2022-11-14

**Authors:** Savvina Moustafa, Katerina Kassela, Maria Bampali, Nikolas Dovrolis, Athanassios Kakkanas, Apostolos Beloukas, Penelope Mavromara, Ioannis Karakasiliotis

**Affiliations:** 1Molecular Virology Laboratory, Department of Microbiology, Hellenic Pasteur Institute, 11521 Athens, Greece; 2Laboratory of Biology, Department of Medicine, Democritus University of Thrace, 68100 Alexandroupolis, Greece; 3National AIDS Reference Center of Southern Greece, Department of Public Health Policy, University of West Attica, 12243 Athens, Greece; 4Molecular Microbiology & Immunology Lab, Department of Biomedical Sciences, University of West Attica, 11521 Athens, Greece; 5Department of Molecular Biology and Genetics, Democritus University of Thrace, 68100 Alexandroupolis, Greece

**Keywords:** hepatitis C virus, core protein, TNFα

## Abstract

Hepatitis C virus (HCV) core protein is a multifunctional protein that is involved in the proliferation, inflammation, and apoptosis mechanism of hepatocytes. HCV core protein genetic variability has been implicated in various outcomes of HCV pathology and treatment. In the present study, we aimed to analyze the role of the HCV core protein in tumor necrosis factor α (TNFα)-induced death under the viewpoint of HCV genetic variability. Immortalized hepatocytes (IHH), and not the Huh 7.5 hepatoma cell line, stably expressing HCV subtype 4a and HCV subtype 4f core proteins showed that only the HCV 4a core protein could increase sensitivity to TNFα-induced death. Development of two transgenic mice expressing the two different core proteins under the liver-specific promoter of transthyretin (TTR) allowed for the in vivo assessment of the role of the core in TNFα-induced death. Using the TNFα-dependent model of lipopolysaccharide/D-galactosamine (LPS/Dgal), we were able to recapitulate the in vitro results in IHH cells in vivo. Transgenic mice expressing the HCV 4a core protein were more susceptible to the LPS/Dgal model, while mice expressing the HCV 4f core protein had the same susceptibility as their littermate controls. Transcriptome analysis in liver biopsies from these transgenic mice gave insights into HCV core molecular pathogenesis while linking HCV core protein genetic variability to differential pathology in vivo.

## 1. Introduction

Hepatitis C virus (HCV) imposes a significant health burden, infecting an estimated 71.1 million individuals worldwide [1]. HCV is responsible for acute and chronic liver disease that often leads to fibrosis/cirrhosis, liver failure, and hepatocellular carcinoma (HCC). The development of potent interferon-free treatments has reduced the incidence of HCV-related liver disease and mortality [2,3]. However, a significant percentage of patients with advanced liver disease will develop HCC, in spite of treatment-induced viral clearance [4].

HCV is a single-stranded positive-sense RNA virus of the family *Flaviviridae*. HCV RNA encodes a polyprotein which is cleaved by host and viral proteases into three structural proteins (the core, E1, and E2 proteins) and seven nonstructural proteins (p7, NS2, NS3, NS4A, NS4B, NS5A, and NS5B) [5,6,7,8,9]. The viral RNA is enclosed in an icosahedral capsid formed by the core protein. This capsid is surrounded by a membrane envelope consisting of E1 and E2 glycoprotein heterodimers. In addition to its principal structural function, HCV core protein interacts with multiple host proteins and, therefore, has a pivotal role in the modulation of pathways such as Toll-like receptor 2 (TLR2), p53, Wnt, interferon, and tumor necrosis factor α (TNFα) signaling [10,11,12,13].

A substantial percentage (25–35%) of patients with cirrhosis have an increased probability of developing bacterial infection that can lead to death [14]. In patients with cirrhosis, the presence of endotoxins induces a systemic inflammatory reaction, with high levels of circulating pro-inflammatory cytokines, which leads to organ failure (acute-on-chronic liver failure) and septic shock [14]. TNFα is a pro-inflammatory cytokine with a key role in many diseases [15]. TNFα acts, through its receptors TNFR1 and TNFR2, on target cells by activating apoptosis pathways. Chronic HCV infection is associated with increased levels of TNFα [16], and elevated TNFα levels are associated with increased liver pathology from fatty liver to hepatocellular carcinoma [17,18]. The HCV core protein is the main viral protein that has been associated with the modification of the cell response to TNFα but also other ligands of the TNF superfamily such as FasL and TRAIL [19,20,21]. However, conflicting reports on the role of HCV core protein have been published. Reports using HCV from various isolates have reported either enhancement of inhibition of TNFα-induced death or other TNF superfamily members [19,20,21,22,23,24,25,26]. Several HCV core proteins were found to interact with the TNFR1 receptor [22], to enhance apoptotic signaling through FADD [23], and to block the survival factor NF-κB [20]. In support of this end, HCV core protein-enhanced TRAIL (TNFSF10) signaling through the activation of Bid was proposed [19]. Other reports claim that the core may inhibit death by TNFα [24,25] through enhancement of c-FLIP [26] and to inhibit apoptosis via the Fas receptor (TNFRSF6) by enhancing the expression of Bcl-xL [21].

The HCV genetic variability is known to affect pathology and therapy [27,28], while polymorphisms in the HCV core protein have been reported to affect steatosis, carcinogenesis, insulin resistance, and interferon response [29]. In the present report, we aimed to investigate the role of the HCV core protein’s genetic variability in the TNFα-associated pathogenesis focusing on the effect of two subtypes of the HCV core protein previously known to differentially modulate host pathways in vitro. Our study highlights the role of HCV genetic variability and host cell specificity of TNFα signaling in an attempt to explain the bibliographic discrepancies.

## 2. Materials and Methods

### 2.1. Cell Lines

Huh-7.5 (kindly provided by Dr. Charles Rice, The Rockefeller University, New York City, NY, USA), immortalized human hepatocytes (IHH, kindly provided by Dr. Ulrike Protzer, Technische Universität München, Munich, Germany), and HEK293T cells (ATCC) were cultured in Dulbecco’s modified Eagle medium (Thermo Fisher Scientific, Waltham, MA, USA), supplemented with non-essential amino acids, 2 mM L-glutamine (Thermo Fisher Scientific, Waltham, MA, USA), 100 µg/mL of penicillin/streptomycin (Thermo Fisher Scientific, Waltham, MA, USA), and 10% fetal bovine serum at 37 °C and 5% CO_2_.

### 2.2. Lentivirus Expression Vectors and Stable Cell Lines

Core4a and core4f genes (Appendix A) were amplified by PCR and cloned into the lentiviral vector pWPI-BLR [30], a derivative of the bicistronic lentiviral vector pWPI. In this vector, the expression of the gene is directed by the human elongation factor 1 alpha (EF1-α) promoter. Blasticidin resistance allows for selection in mammalian cells. Lentivirus particles were produced in 293T cells using packaging constructs pCMVR8.91 and pMD.G, as previously described [31]. Infection of Huh7.5 and IHH cells with lentivirus particles and selection with blasticidin yielded the respective Huh7.5-core4a, Huh7.5-core4f, IHH-core4a, and IHH-core4f stable cell lines. Lentivirus particles produced with empty pWPI-BLR vectors were used for the production of the control stable cell lines Huh7.5-C and IHH-C. Western blot analysis was performed using anti-GAPDH (6C5, Abcam, Cambridge, MA, USA) and anti-HCV core (C7-50, Abcam, Cambridge, MA, USA) antibodies according to the manufacturer’s general guidelines. 

### 2.3. TNF-Cycloheximide Cytotoxicity Assay

TNF-cycloheximide (CHX) cytotoxicity assays were carried out on both Huh and IHH derived stable cell lines. Cells were plated at a 10^5^ density in a 96-well plate and cultured for 24 h in Dulbecco’s modified Eagle medium (Thermo Fisher Scientific, Waltham, MA, USA) and supplemented with non-essential amino acids, 2 mM L-glutamine (Thermo Fisher Scientific, Waltham, MA, USA), 100 µg/mL of penicillin/streptomycin (Thermo Fisher Scientific, USA), and 10% fetal bovine serum at 37 °C and 5% CO_2_. Cells were treated with 2 μg/mL CHX (Cell Signaling Technology, Danvers, MA, USA) and a range of human TNFα (Peprotech, Rocky Hill, NJ, USA) concentrations and were incubated at 37 °C and 5% CO_2_ for 18 h. MTT assay was carried out, as previously described [31].

### 2.4. UV Irradiation Assays

UV irradiation assays were carried out on IHH derived stable cell lines. Cells were plated at a 10^5^ density in a 96-well plate and irradiated using UV-C at 1 J/m^2^ s for a range of time periods in a UV crosslinker (Stratagene, Santa Clara, CA, USA). Cells were subsequently cultured for 24 h in Dulbecco’s modified Eagle medium (Thermo Fisher Scientific, USA) and supplemented with non-essential amino acids, 2 mM L-glutamine (Thermo Fisher Scientific), 100 µg/mL of penicillin/streptomycin (Thermo Fisher Scientific, USA), and 10% fetal bovine serum at 37 °C and 5% CO_2_. MTT assay was carried out as previously described [31].

### 2.5. Transgenic Mice

HCV core 4a and core 4f genes were amplified by PCR from pCI/core-4aR, pCI/core-4fC plasmid (https://www.ncbi.nlm.nih.gov/pmc/articles/PMC6060129/) (accessed on 12 April 2022) and placed downstream of a transthyretin (TTR) liver-specific promoter [32]. Core 4a and core 4f genes was inserted into the *Stu*I site of the pTTR1-ExV3 plasmid, and the transgenes were prepared by purifying the *Hind*III fragment containing TTR promoter and the core 4a/core 4f coding sequence. The TTR promoter was kindly provided by Dr. Iannis Talianidis, Institute of Molecular Biology and Biotechnology of FORTH in Crete, Heraklion, Greece. Transgenic CBA-C57BL/6 mice, which express liver-specifically core 4a (TTRcore4a) and core 4f (TTRcore4f) proteins, were generated in the Transgenesis Facility of the Biomedical Sciences Research Center “Alexander Fleming”, Vari, Greece. Mice were maintained under specific pathogen-free (SPF) conditions. Western blot using anti-GAPDH (6C5, Abcam, Cambridge, MA, USA) and anti-HCV core (C7-50, Abcam, Cambridge, MA, USA) antibodies was conducted according to the manufacturer’s general guidelines.

### 2.6. Murine Model of Acute Liver Failure

Eight-week-old male mice were administered Lipopolysaccharide/D-galactosamine (LPS/Dgal), and their acute liver failure was plotted in a Kaplan-Meier survival curve, as described previously [33]. Statistical comparison was conducted by the Log-rank (Mantel-Cox) test using default parameters in GraphPad Prism 8.02 (Dotmatics, Boston, MA, USA).

### 2.7. RNA Sequencing and Bioinformatics Analysis

Total RNA was extracted using Trizol (Thermo Fisher Scientific, Waltham, MA, USA) from biopsies of the large liver lobe of TTR-core4a, TTR-core4f transgenic mice and their littermate wild-type (wt) controls. Total RNA was sequenced using the 3′RNA sequencing (3′RNAseq) protocol (Lexogen, Vienna, Wien, Austria) on an Ion Proton sequencer at the Genomics Facility of the Biomedical Sciences Research Center “Alexander Fleming”, Vari, Greece. Raw 3′RNAseq reads were aligned on the Mus_musculus.GRCm38.96 reference murine transcriptome using salmon v1.6.0 [34] and its quasi-mapping capabilities. Read counts were imported into a custom R script using tximport v 1.20.0 [35] and provided as input for differential gene expression analysis (DGEA) via DESeq2 v1.32.0 [36]. DESeq2’s plotPCA function was also used to create the PCA plot. Gene annotation mappings from Ensembl mus musculus gene identifiers to murine and human entrez gene identifiers were performed using the org.Mm.eg.db [37], org.Hs.eg.db [38], and R packages. Overrepresentation pathway analysis and visualizations were performed via the Reactome [39] database using the clusterProfiler v4.0.5 [40] R package on the genes which were found differentially expressed with a fold regulation of ±2 and adjusted −*p* < 0.01. clusterProfiler was also used for the creation of the gene–concept networks linking Reactome pathways to their respective genes (cnetplots). Finally, EnhancedVolcano v1.10.0 [41] was used in the creation of the volcano plot for the differentially expressed genes from 3′RNAseq. Datasets are available at BSRC “Alexander Fleming” genomic repository (https://genomics-lab.fleming.gr/fleming/external/Karakasiliotis/run342/metaseqr_quantseq_run342a/index.html) (accessed on 20 November 2020).

### 2.8. Real-Time RT-PCR

Quantitative real-time reverse transcription (RT) polymerase chain reaction (PCR) was conducted using cDNA from RNA extracted from liver biopsies. Two μg of RNA were used as a template for reverse transcription by murine leukemia virus reverse transcriptase (MLV RT) (Promega, Madison, WI, USA) with oligo d(T)s at 42 °C for 60 min. Real-time PCR was performed in a Mx3005P Real-Time PCR System (Agilent, Santa Clara, CA, USA) using KAPA SYBR FAST qPCR Master Mix (2X) Kit (Sigma-Aldrich, St. Louis, MO, USA) following the manufacturer’s protocol and custom oligonucleotide primers (Appendix A). The conditions used for cycling were the following: 40 cycles of 95 °C for 10 s, 60 °C for 20 s, and 72 °C for 15 s. Relative mRNA expression was calculated using the ΔΔCt method [42] and GAPDH mRNA as a normalizer. The two-tailed *t*-test was used for *p*-value calculation using default parameters in GraphPad Prism 8.02 (Dotmatics, Boston, MA, USA).

## 3. Results

### 3.1. HCV Core Presents Differential Susseptibility to TNFα-Induced Death In Vitro

HCV core proteins from two clinical isolates, corresponding to subtypes 4a and 4f, previously isolated in Romania and Cameroon, were stably expressed in Huh7.5 and IHH cell lines [43]. The pWI-BLR vector was used for cloning the core 4a and core 4f genes and the respective lentivirus particles production (Figure 1a). Core 4a and core 4f stably expressing Huh7.5 hepatoma cell lines (Figure 1b), treated with TNFα and cycloheximide, showed similar death responses when compared to control cells (Figure 1c). As Huh7.5 cells may have deregulated death-related pathways, we used a non-cancerous cell line; the immortalized human hepatocytes (IHH). IHH cells were tested for their hepatic origin using the AT1A HNF4A mRNA markers as compared to Huh7.5 and 293T cells (Figure 1d). In IHH cells, the core 4a protein resulted in a significant reduction of cell viability at various concentrations of TNFα, while core 4f showed minimal effect on cell viability (Figure 1e). Thus, core 4a and core 4f proteins not only presented differential modulation of TNFα pathway, but also this effect was cell line specific. To assess the specificity of this result, we tested the IHH stable cell lines in a different model of apoptosis: the UV-induced apoptosis. UV irradiation of IHH 4a and IHH 4f cells resulted in similar levels of cell death between cell lines and as compared to IHH control cells (Figure 1f). Similar levels of UV-induced death may signify that the apoptosis pathway where TNFα-induced apoptosis and UV-induced apoptosis converge is not differentially affected by the two types of core protein.

### 3.2. HCV Core Expression Results in Differential Susseptibility to the LPS/Dgal Hepatic Failure Model

As previously described, the role of the HCV core protein in the TNFα pathway in various cell lines is controversial. Thus, we aimed for an in vivo animal model using transgenic mice that express HCV core proteins. A TTR-promoter-driven liver-specific expression of HCV core 4a and HCV core 4f proteins (Figure 2a) allowed us to assess the in vivo role of the two subtypes of core protein. TTRcore4a and TTRcore4f transgenic mice were generated using pronuclear injection, and core protein expression was assessed using Western blot (Figure 2b), while HCV core mRNA expression was measured in various tissues to verify tissue specificity (Appendix A). The transgene for both lines was inherited at the expected 1:1 ratio (Appendix A).

A well studied in vivo mouse model for liver pathology is LPS/Dgal acute liver failure. The model is based on TNF-induced death of hepatocytes after LPS administration and D-galactosamine hepatocyte death sensitization. TTRcore4a mice showed increased susceptibility to LPS/Dgal treatment as compared to littermate control (Figure 2c). The result was verified using a second founder line bearing the same transgene. On the other hand, TTRcore4f mice showed similar susceptibility to LPS/Dgal treatment as compared to littermate control. Although, at later timepoints, a non-significant increase in death was observed (Figure 2c). The result was verified using a second founder line bearing the same transgene.

### 3.3. Subtype-Specific Modulation of Molecular Pathways by HCV Core In Vivo

The difference in the response to LPS/Dgal between TTRcore4a and TTRcore4f transgenic mice was investigated through differential transcriptome analysis using 3′RNAseq. Differential transcriptome analysis of TTRcore4a (Table 1) and TTRcore4f (Table 2), compared to their littermate controls, yielded a small number of upregulated and downregulated genes using a cut-off of FDR < 0.05 and fold change >2 or <−2. The resulted differentially expressed genes did not fall into a statistically significant gene set after gene set enrichment analysis (www.gsea-msigdb.org, accessed on 12 April 2022). However, individual genes that have been in the past implicated in liver pathology were identified (Table 1 and Table 2). These mRNAs were quantified using real-time RT-PCR validating the bioinformatics analysis (Figure 3). SAA1 and SAA2 mRNAs were significantly upregulated in TTRcore4a mice compared to their littermate wt controls, while ADAMDEC1, MT1, MT2, HAMP2, and GSN mRNAs were downregulated. SAA2 mRNA was significantly upregulated in TTRcore4f mice compared to their littermate wt controls, while no mRNAs significant to liver pathology were found downregulated.

To further explore the pathogenetic mechanism, we further explored the differential expression dataset for genes regulated downstream of SAA1, SAA2, ADAMDEC1, MT1, MT2, and GSN or related to cell death and liver proliferation. Because SAA1 overexpression has shown to increase various cytokines [44], we assessed the expression of CCL2, CCL3, CCL4, CCL7, and CCL11. The mRNA expression for 5 cytokines was found significantly upregulated in TTRcore4a mice (Figure 3). Accordingly, the relative abundance of a specific T-cell marker IL2RG was significantly increased in TTRcore4a mouse livers (Figure 3). While proinflammatory cytokines were upregulated, we assessed the expression of TNFα related mRNAs, such as TNFα, TNFR1, TNFR2, and cFLIP. TNFα, TNFR1, and TNFR2 mRNAs were expressed at similar levels (Figure 3) in both mouse lines, while cFLIP was overexpressed in TTRcore4f mice, reflecting a possible anti-apoptotic potential.

## 4. Discussion

Recent advances in direct acting antiviral development have revolutionized the therapy of HCV infection. However, a sustained virological response in a significant proportion of patients does not reverse liver pathology [45,46]. Pathological mechanisms in liver fibrosis and cirrhosis in HCV patients [47,48], and in vivo models [49], have mainly highlighted the deregulation of inflammatory and metabolic processes. The HCV core protein is central in the host-pathogen interaction network in HCV pathology, involved in inflammatory and metabolic aspects. The HCV core protein interacts with cellular proto-oncogenes and changes their expression patterns, resulting to hepatocarcinogenesis [50,51]. In vitro, HCV core protein activated interleukin 6 (IL-6) and interleukin 8 (IL-8) through TLR2 while activating NF-κB in parallel [52,53]. Stimulation of naïve human macrophages with HCV core induces TNF-α and IL-6 production through the TLR2 pathway [10]. HCV core binding to transcription factor signal transducer and activator of transcription 1 (STAT1) appears to inhibit the activation of innate immunity through the type I interferon pathway [11]. Downstream of p53, HCV core interacts with p21WAF that negatively regulates the kinase/cyclin system, consequently accelerating the cell cycle [12,13]. The development of transgenic mice that expressed the HCV core protein in a liver-specific manner led to steatosis, a clinical feature that is characteristic of patients with chronic HCV infection [54,55]. These mice, at a later stage in their lives, spontaneously presented hepatocellular carcinoma [55,56,57]. Moreover, crossing HCV core transgenic mice with PPARα^−/−^ mice indicated that the HCV core protein is implicated in the activation of PPARα, a nuclear receptor that has an essential role in lipid homeostasis [57].

TNFα has a central role in liver inflammation [58] and metabolic [59] processes, and it constitutes a hallmark in liver fibrosis [60]. However, conflicting published results did not allow for a definitive role of HCV core protein in TNFα signaling. On one hand, the HCV core was found to interact with the TNFR1 receptor (HCV genotype 1b in Huh-7 cells) [22], to enhance apoptotic signaling through FADD (HCV genotype 1b in Huh-7 cells) [23], and to block the survival factor NF-κB (HCV genotype 2a JFH1 clone in Huh-7.5 cells) [20]. Accordingly, another report exemplified HCV core enhanced TRAIL signaling through increased activation of Bid and the mitochondrial branch of apoptosis (HCV genotype 1b in Huh-7 cells) [19]. In contrast, the HCV core was found to inhibit death by TNFα (HCV genotype 1a in MCF7 cells) [24,25] through the enhancement of c-FLIP (HCV genotype 1a in HepG2 cells) [26] and to inhibit Fas-induced apoptosis through enhancement of Bcl-xL (HCV genotype 1b J4L6S clone in HepG2 cells) [21]. Considering the impact of HCV genetic variability on HCV pathogenesis [29], it is likely that the observed discrepancies are actually the effect of polymorphisms within the core protein of various genotypes/strains used in the above studies.

In order to assess the role of HCV genetic variability on TNFα-induced death, we used core proteins from two subtypes of HCV genotype 4, namely, 4a and 4f. These two core type have been recently shown to differentially modulate the Wnt pathway in 293T and Huh7.5 cells [43]. The HCV 4f core protein enhanced Wnt signaling more than the HCV 4a core protein. Interestingly, an amino acid substitution (S71T) in core 4a was able to recapitulate the effect of core 4f, highlighting the role of the specific amino acid in Wnt pathway regulation [43]. The interplay of Wnt and TNFα signaling has been well documented [61,62,63,64]. HCV core 4a and core 4f proteins did not present differential susceptibility to TNFα-induced death in Huh7.5, the cell line that is often used for HCV-related research. As Huh7.5 cells present a strong epithelial to mesenchymal transition (EMT) phenotype [65], and have accumulated several mutations regarding apoptosis related genes such as *BAX*, *MAP3K1*, and *TP53* [66], we used an immortalized human hepatocyte cell line (IHH). The IHH core 4a protein induced higher susceptibility to TNFα-induced death, highlighting the importance not only of HCV genetic variability but, also, the importance of the cell line genetic background and phenotype.

Mouse models have been at the forefront of the understanding of the role of the HCV core protein in liver pathology, mainly dissecting the steatosis–hepatocarcinogenesis axis [49]. With the development of two novel core-protein-expressing mouse transgenic lines, TTRcore4a and TTRcore4f, we aimed for the recapitulation of TNFα susceptibility observed in IHH cells. The LPS/Dgal-liver-failure mouse model is well known to be dependent on TNFα. The model is based on TNFα production after LPS treatment and subsequent TNFα-induced apoptosis, as TNFα and tumor necrosis factor receptor 1 (TNFR1) knockout mice showed no liver pathology [67]. TTRcore4a mice showed enhanced susceptibility to the LPS/Dgal model, while TTRcore4f mice showed similar susceptibility to their littermate controls. The effect was similar to the effect presented in IHH cell lines stably expressing core 4a and core 4f. Thus, HCV subtype 4a and 4f core proteins seem to present a differential effect on liver pathology, further supporting the role of HCV core protein variability in a genotype-specific pathology. As the amino acid sequence of the two strains were different only in two positions, and the amino acid 191 is cleaved off in cells by signal peptidase [68], the presented effect may be attributed to position 71 that has been, in the past, implicated in various pathological mechanisms.

Analysis of total mRNA transcriptome in the TTRcore4a and TTRcore4f transgenic resulted in a specific small set of differentially expressed mRNAs. Although pathway analysis did not yield statistically significant gene groups, some key mRNAs in liver pathology were identified and further analyzed using real time RT-PCR. One of the most downregulated mRNAs in TTRcore4a livers was ADAM-like DECysin-1 (*ADAMDEC1*), which is a secreted metalloprotease [69] that is highly associated with the gastrointestinal tract. Adamdec1^−/−^ mice were more susceptible to the induction of bacterial and dextran sodium sulphate (DSS) induced colitis, and they presented increased bacteremia [70]. Gelsolin (Gsn) was one of the mRNAs found downregulated only in TTRcore4a mice. Gsn deficient mice presented increased susceptibility to the anti-Fas model of liver failure [71], a model that is based on similar signaling pathways to LPS/Dgal model, as Fas receptor belongs to the TNF receptor superfamily [72]. *SAA1* mRNA, in contrast to *SAA2* mRNA, was upregulated only in TTRcore4a mice. In previous reports, hepatic overexpression of *SAA1* aggravated fatty liver inflammation by promoting intrahepatic platelet aggregation [73], while, recently, *SAA1* was shown to exacerbate hepatic steatosis via the TLR4-mediated NF-κB signaling pathway [74]. *SAA1* overexpression may lead to the induction of pro-inflammatory genes as acute response mediators during infection [75], and they aggravated T cell-mediated hepatitis in mice [44]. The mRNA of Mt1 and Mt2 metalotheionines were found downregulated in both transgenic mice and, thus, may not account for the differential response to LPS/Dgal, although they are highly associated with the response to HCV infection [76]. Finally, another mRNA associated with liver function was that of hepcidin (*HAMP*) and hepcidin 2 (*HAMP2*), which are mouse-specific paralogs of *HAMP*. *HAMP* was minimally affected in both transgenic mice, while *HAMP2* cannot be linked to HCV infection, as it is mouse-specific. However, combined with the effect on *ADAMDEC1* and metallothionines, expression may pinpoint a deregulation in metal metabolism in the core-expressing liver, a hallmark in liver pathology [77]. The differential expression analysis highlighted several mediators of pathogenesis that require validation in human biopsies and may shed light into hepatocyte death during HCV infection, directly associated with liver fibrosis and cirrhosis.

Further analysis of the presented phenotype signified differences in the inflammatory status of the livers of TTRcore4a mice. Thus, we further analyzed inflammation effectors and modulators, such as the cytokines shown before, to aggravate T cell-mediated hepatitis following *SAA1* overexpression [44]. Indeed, we observed an increased expression of *CCL2*, *CCL3*, *CCL7*, and *CCL11* in TTRcore4a mouse livers, while a statistically significant increase in the presence of IL2RG marker was also observed. Expression of *TNFα* mRNA and the mRNAs of its receptors *TNFR1* and *TNFR2* mRNAs remained the same between transgenic mice and their wild-type controls. This pro-inflammatory environment signifies a sensitized background towards liver inflammation that possibly aggravates the phenotype in the LPS/DGAL model, similarly to the T cell-mediated hepatitis model published previously [44]. As *cFLIP* is a direct link between the inflammatory status, TNF-induced apoptosis [78,79], and the HCV core protein pathogenesis [26], we assessed the role of HCV core in the expression of *cFLIP* in vivo. Intriguingly, cFLIP was significantly induced in the livers of TTRcore4f mice, possibly contributing to the observed difference between TTRcore4a and TTRcore4f mice in the LPS/DGAL model.

Overall, in this study, we identified an aggravating/sensitizing role of HCV core protein in TNFα-induced apoptosis in both in vitro and in vivo models. This role was determined by the genetic variability of the HCV core protein and the background of the mammalian cells used and, thus, our study offers an alternative point of view to the conflicting reports of current bibliography on TNFα-induced apoptosis. Our findings should be validated in patients and infectious systems. Despite the fact that overexpression models have shed light onto core protein pathogenesis, they still remain artificial systems [49,80].

## Figures and Tables

**Figure 1 viruses-14-02521-f001:**
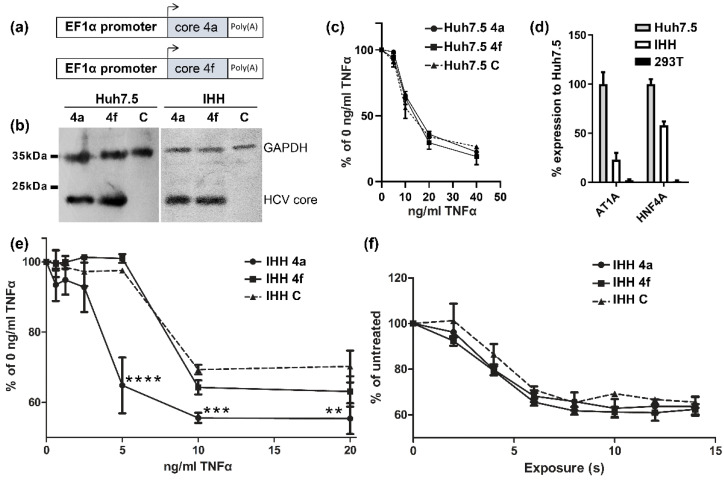
Assessment of HCV core protein role in the modulation of TNFα-induced death in vitro. (**a**) Schematic representation of the lentivirus expression construct used for generation of the stable Huh7.5 and IHH cell lines. (**b**) Western blot of Huh7.5 and IHH cell lines stably expressing HCV core 4a and 4f. (**c**) TNFα/CHX toxicity assay in Huh7.5 cells stably expressing HCV core 4a and 4f compared to control Huh7.5 cells. (**d**) Real-time RT-PCR for the quantification of AT1A and HNF4A mRNAs in Huh7.5, IHH, and negative control 293T cells. (**e**) TNFα/CHX toxicity assay in IHH cells stably expressing HCV core 4a and 4f compared to control IHH cells. (**f**) UV-induced death of IHH cells stably expressing HCV core 4a and 4f proteins compared to control IHH cells. (** *p* < 0.02, *** *p* < 0.01, **** *p* < 0.001).

**Figure 2 viruses-14-02521-f002:**
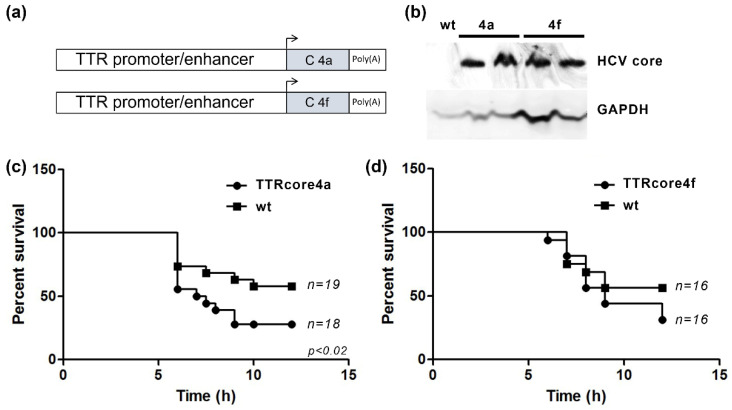
Assessment of HCV core role in a TNFα-depended hepatic failure in vivo model. (**a**) Schematic representation of the transgene construct used for generation of the TTRcore4a and TTRcore4f transgenic mice. (**b**) Western blot of total protein from liver biopsies of TTRcore4a and TTRcore4f mice and a wt control (two representative mice were used for each line). (**c**) Kaplan-Meier survival curves for TTRcore4a male mice and the respective littermate controls after administration of LPS/Dgal. At the bottom right corner, the *p* value is reported. (**d**) Kaplan-Meier survival curves for TTRcore4f male mice and the respective littermate controls after administration of LPS/Dgal.

**Figure 3 viruses-14-02521-f003:**
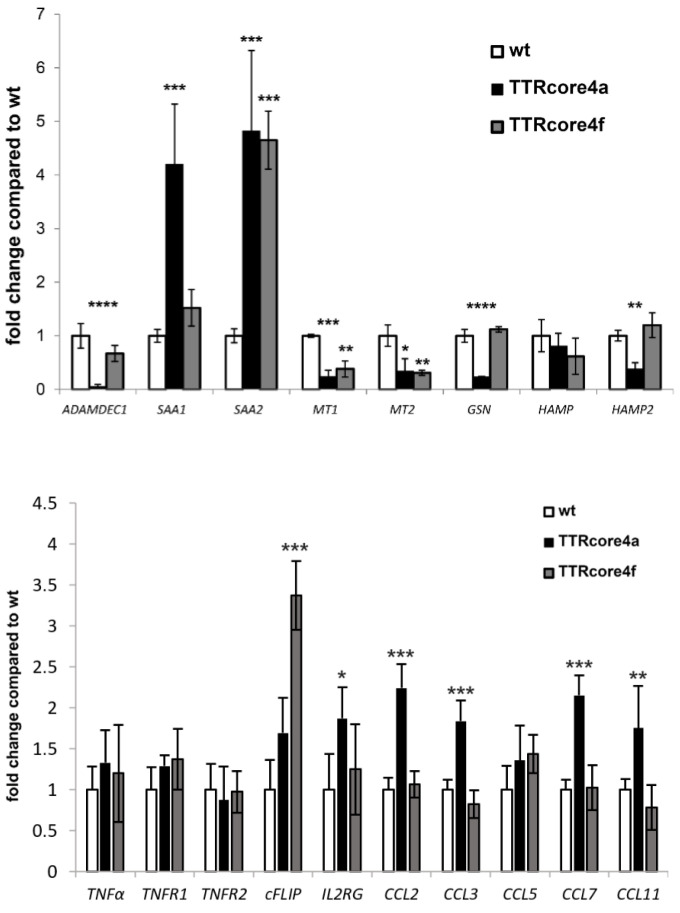
Real-time RT–PCR validation of differentially expressed genes in the liver of TTRcore4a and TTRcore4f transgenic mice. Fold change to the wt littermate controls are depicted in the barchart. Error bars correspond to the standard deviation from three biological replicates. (* *p* < 0.05, ** *p* < 0.02, *** *p* < 0.01, **** *p* < 0.001).

**Table 1 viruses-14-02521-t001:** Differentially expressed genes between TTRcore4a transgenic mice and the respective littermate wt controls.

Gene Symbol	*p*-Value	FDR	Fold Change
Upregulated Genes			
*HHLA1*	0.0000000	0.0000000	53.00
*TNNC2*	0.0000000	0.0000091	21.67
*ACTA1*	0.0000056	0.0045909	11.00
*SERPINE1*	0.0000416	0.0195843	9.33
*SAA1*	0.0000000	0.0000441	5.96
*SAA2*	0.0000000	0.0000762	5.82
*GM42674*	0.0000206	0.0107402	4.96
*S100A4*	0.0000206	0.0107402	4.96
*SELENBP2*	0.0000013	0.0012631	3.79
*MUP17*	0.0000011	0.0011299	3.58
*MUP7*	0.0000077	0.0059425	3.55
*MUP15*	0.0000297	0.0144767	3.39
*CYP7B1*	0.0000017	0.0015311	3.12
*HSP90AA1*	0.0000236	0.0119081	2.96
*MUP11*	0.0001180	0.0462593	2.91
*TSKU*	0.0000117	0.0075599	2.90
*BAG3*	0.0000853	0.0366297	2.61
Downregulated Genes			
*ADAMDEC1*	0.0000000	0.0000000	−23.32
*CLEC3B*	0.0000000	0.0000079	−15.50
*INO80DOS*	0.0000651	0.0296855	−13.13
*FBLN1*	0.0000003	0.0003058	−8.59
*CRISPLD2*	0.0000000	0.0000348	−7.64
*HTRA3*	0.0000133	0.0080787	−6.45
*CYP26A1*	0.0000007	0.0007748	−6.07
*MATN2*	0.0001204	0.0462593	−6.00
*LPAR1*	0.0000098	0.0071805	−5.91
*ELN*	0.0001141	0.0462593	−5.18
*GSN*	0.0000000	0.0000011	−5.18
*NIPAL1*	0.0000199	0.0107402	−4.64
*HAMP2*	0.0000000	0.0000004	−4.57
*DPT*	0.0000119	0.0075599	−4.56
*COL6A3*	0.0000205	0.0107402	−3.58
*MGP*	0.0000057	0.0045909	−3.44
*MT2*	0.0000001	0.0001359	−3.20
*AQP8*	0.0000001	0.0001509	−3.14
*CYP4A14*	0.0000719	0.0318185	−2.96
*COL1A1*	0.0000922	0.0384517	−2.78
*MT1*	0.0000104	0.0072522	−2.52
*8430408G22RIK*	0.0001333	0.0499041	−2.06

**Table 2 viruses-14-02521-t002:** Differentially expressed genes between TTRcore4f transgenic mice and the respective littermate wt controls.

Gene Symbol	*p*-Value	FDR	Fold Change
Upregulated Genes			
*IKBIP*	0.0000023	0.0033358	8.10
*SAA2*	0.0000003	0.0005697	5.85
*HAMP*	0.0000612	0.0470078	3.29
Downregulated Genes			
*LOX*	0.0000000	0.0000010	−21.19
*ANKRD1*	0.0000165	0.0185512	−14.10
*DMBT1*	0.0000000	0.0000000	−12.44
*S100A4*	0.0000000	0.0000000	−10.19
*GM42674*	0.0000000	0.0000000	−10.05
*RP24-361O20.1*	0.0000005	0.0008336	−9.67
*GM16198*	0.0000372	0.0339678	−7.80
*DDIT4*	0.0000000	0.0000053	−6.77
*S100A6*	0.0000000	0.0000013	−6.09
*GM20649*	0.0000145	0.0176227	−5.42
*MUP-PS17*	0.0000002	0.0003379	−5.14
*MEG3*	0.0000433	0.0351342	−3.97
*DNAAF5*	0.0000326	0.0317683	−3.95
*EIF4G1*	0.0000040	0.0053495	−3.71
*LCOR*	0.0000178	0.0185512	−3.57
*CYP26B1*	0.0000404	0.0347101	−2.91

## Data Availability

Total 3′RNA sequencing datasets are available at the BSRC “Alexander Fleming” genomic repository (https://genomics-lab.fleming.gr/fleming/external/Karakasiliotis/run342/metaseqr_quantseq_run342a/index.html) (accessed on 20 November 2020).

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
