# Peer review of "Hepatitis C Core Protein Induces a Genotype-Specific Susceptibility of Hepatocytes to TNF-Induced Death In Vitro and In Vivo"

_viruses, 2022, doi:10.3390/v14112521_

Round 1
Reviewer 1 Report
Moustafa et al. examined the effect of over-expressing HCV Core derived from genotype 4a and 4f in vitro and in vivo. While 4a and 4f core do not affect toxicity of TNFa in Huh7.5 cells, 4a sensitize cells to the effect of TNFa in IHH and transgenic mice. Subsequent transcriptome analysis showed 4a core specifically regulate a subset of genes compared to 4f core. The different expression of these genes might be associated with the enhanced TNFa-dependent cell death sensitivity.
The data supports the idea that the HCV core derived from different isolates has differential cellular effects. Experiments have included appropriate controls to support this conclusion. Although the paper presentsinteresting initial observations, it can be benefited from including further experimentations to understand the mechanism of action. For example, what is the effect of overexpression of the 4a or 4f core on different factors along the TNFa signaling pathway? Furthermore, it is hard to connect how over-expression of core reported in this study compared to the actual HCV infection, i.e. Is there any different pathology between patients infected with HCV 4a and 4f?
Furthermore, there are two minor points:
1) In figure 1f. Please double check the p-values reported. It seems to be backward where p-values seem to be smaller in data points that have more overlap.
2) In line 314, since S71T played a role in WNT pathway. Would the mutation similarly affect the TNFa sensitivity reported here?
Author Response
Reviewer 1
Moustafa et al. examined the effect of over-expressing HCV Core derived from genotype 4a and 4f in vitro and in vivo. While 4a and 4f core do not affect toxicity of TNFa in Huh7.5 cells, 4a sensitize cells to the effect of TNFa in IHH and transgenic mice. Subsequent transcriptome analysis showed 4a core specifically regulate a subset of genes compared to 4f core. The different expression of these genes might be associated with the enhanced TNFa-dependent cell death sensitivity.
-We would like to thank the reviewers for taking the time to review our manuscript.
The data supports the idea that the HCV core derived from different isolates has differential cellular effects. Experiments have included appropriate controls to support this conclusion. Although the paper presents interesting initial observations, it can be benefited from including further experimentations to understand the mechanism of action. For example, what is the effect of overexpression of the 4a or 4f core on different factors along the TNFa signaling pathway? Furthermore, it is hard to connect how over-expression of core reported in this study compared to the actual HCV infection, i.e. Is there any different pathology between patients infected with HCV 4a and 4f?
-We have conducted further experiments adding more mRNAs in the analysis, most of them either related to TNF or related to inflammatory status in general. Analysis of core overexpression in mice is the main tool the scientific community has used in order to assess in vivo the role of the HCV core protein.
Related recent publications
1: Jia F, Diao P, Wang X, Hu X, Kimura T, Nakamuta M, Nakamura I, Shirotori S,
Sato Y, Moriya K, Koike K, Gonzalez FJ, Nakayama J, Aoyama T, Tanaka N. Dietary
Restriction Suppresses Steatosis-Associated Hepatic Tumorigenesis in Hepatitis C
Virus Core Gene Transgenic Mice. Liver Cancer. 2020 Sep;9(5):529-548. doi:
10.1159/000508308. Epub 2020 Jul 10. PMID: 33083279; PMCID: PMC7548900.
2: Diao P, Wang X, Jia F, Kimura T, Hu X, Shirotori S, Nakamura I, Sato Y,
Nakayama J, Moriya K, Koike K, Gonzalez FJ, Aoyama T, Tanaka N. A saturated
fatty acid-rich diet enhances hepatic lipogenesis and tumorigenesis in HCV core
gene transgenic mice. J Nutr Biochem. 2020 Nov;85:108460. doi:
10.1016/j.jnutbio.2020.108460. Epub 2020 Jul 3. PMID: 32992072; PMCID:
PMC7756930.
3: Hu X, Wang X, Jia F, Tanaka N, Kimura T, Nakajima T, Sato Y, Moriya K, Koike
K, Gonzalez FJ, Nakayama J, Aoyama T. A trans-fatty acid-rich diet promotes
liver tumorigenesis in HCV core gene transgenic mice. Carcinogenesis. 2020 Apr
22;41(2):159-170. doi: 10.1093/carcin/bgz132. PMID: 31300810; PMCID: PMC8456504.
4: Zheng Y, Shimamoto S, Maruno T, Kobayashi Y, Matsuura Y, Kawahara K, Yoshida
T, Ohkubo T. N-terminal HCV core protein fragment decreases 20S proteasome
activity in the presence of PA28γ. Biochem Biophys Res Commun. 2019 Feb
5;509(2):590-595. doi: 10.1016/j.bbrc.2018.12.167. Epub 2018 Dec 31. PMID:
30602418.
5: Abdallah C, Lejamtel C, Benzoubir N, Battaglia S, Sidahmed-Adrar N, Desterke
C, Lemasson M, Rosenberg AR, Samuel D, Bréchot C, Pflieger D, Le Naour F,
Bourgeade MF. Hepatitis C virus core protein targets 4E-BP1 expression and
phosphorylation and potentiates Myc-induced liver carcinogenesis in transgenic
mice. Oncotarget. 2017 Apr 20;8(34):56228-56242. doi: 10.18632/oncotarget.17280.
PMID: 28915586; PMCID: PMC5593557.
6: Sekine S, Ito K, Watanabe H, Nakano T, Moriya K, Shintani Y, Fujie H,
Tsutsumi T, Miyoshi H, Fujinaga H, Shinzawa S, Koike K, Horie T. Mitochondrial
iron accumulation exacerbates hepatic toxicity caused by hepatitis C virus core
protein. Toxicol Appl Pharmacol. 2015 Feb 1;282(3):237-43. doi:
10.1016/j.taap.2014.12.004. Epub 2014 Dec 27. PMID: 25545986.
7: Noh DH, Lee EJ, Kim AY, Lee EM, Min CW, Kang KK, Lee MM, Kim SH, Sung SE,
Hwang M, Yu DY, Jeong KS. Alcohol induced hepatic degeneration in a hepatitis C
virus core protein transgenic mouse model. Int J Mol Sci. 2014 Mar
7;15(3):4126-41. doi: 10.3390/ijms15034126. PMID: 24608925; PMCID: PMC3975388.
Furthermore, there are two minor points:
1) In figure 1f. Please double check the p-values reported. It seems to be backward where p-values seem to be smaller in data points that have more overlap.
Thank you they were placed wrongly
2) In line 314, since S71T played a role in WNT pathway. Would the mutation similarly affect the TNFa sensitivity reported here?
We added a respective comment in the discussion section
Reviewer 2 Report
There are some issues in this manuscript that need to be fixed.
1. Statistical methods used in the manuscript should be described in the Methods section.
2. N number of mice in Western blot, N ≥3 would be better.
3. Results — 3.1. Indicate the description of Fig.e.d.f in the manuscript.
4. Results — 3.2. (line 214). Where is the figure for Figure S1?
5. The picture of Figure 1. should appear after Result 3.1.
6. Lines 271, 272, (** P <0.05, ** P <0.02) Please specify which one is **.
7. Please unify abbreviations in this manuscript, e.g, SAA, FasL, TP53, IHH, TRAIL.
8. Authors should discuss the limitations of this manuscript and prospects for the future.
9. The authors should explore more in the introduction section.
10. What are the possible mechanisms by which HCV core 4a increases susceptibility for TNFα ?
11. Is it possible that 4a increases the expression of TNFR or the affinity to TNFα? Or do you think that the increased sensitivity to TNFα is mainly due to differences in gene expression between the variants detected in this study?
12. There is a typographical error, (e) is duplicated in Figure 1 ((f) is absent).
Author Response
We would like to thank the reviewer for taking the time to review our manuscript
There are some issues in this manuscript that need to be fixed.
- Statistical methods used in the manuscript should be described in the Methods section.
The statistical methods were added both in survival analysis and differential expression methodologies
- N number of mice in Western blot, N ≥3 would be better.
A supplementary figure was added with all the samples assessed by western blot for 3 or more mice per line. We also included mendelian ratios to show a correct inheritance profile of the transgene.
- Results — 3.1. Indicate the description of Fig.e.d.f in the manuscript.
The corresponding changes were made
- Results — 3.2. (line 214). Where is the figure for Figure S1?
The figure was in the separate supplementary file. We have added one more figure and you can again access it through the system.
- The picture of Figure 1. should appear after Result 3.1.
The corresponding changes were made
- Lines 271, 272, (** P <0.05, ** P <0.02) Please specify which one is **.
The corresponding changes were made
- Please unify abbreviations in this manuscript, e.g, SAA, FasL, TP53, IHH, TRAIL.
We have looked through the manuscript and we did not find any different abbraviations for the same gene/protein. However, only the gene was in italics.
- Authors should discuss the limitations of this manuscript and prospects for the future.
We have added this in the conclusions at the end of the discussion.
- The authors should explore more in the introduction section.
We have added more details on the bibliographic discrepancies over the role of core protein in TNF apoptosis.
- What are the possible mechanisms by which HCV core 4a increases susceptibility for TNFα ?
We have added more on the issue in both results and discussion, exploring more the inflammatory component of the liver.
- Is it possible that 4a increases the expression of TNFR or the affinity to TNFα? Or do you think that the increased sensitivity to TNFα is mainly due to differences in gene expression between the variants detected in this study?
We believe that as core protein is a multifunctional protein there is not only one facet of its role in the in vivo system. Possibly this is why it was hard for us to identify a specific pathway. The balance between extracellular singling such as other cytokines, liver injury and regeneration signals are also critical. However, we believe that contrary to the previous studies that assessed the role of core in the TNF apoptosis in cell culture, our approach may give better insights into it inside a model system liver.
- There is a typographical error, (e) is duplicated in Figure 1 ((f) is absent).
The error was corrected
Reviewer 3 Report
The manuscript entitled “ Hepatitis C core protein induces a genotype-specific susceptibility of hepatocytes to TNF-induced death in vitro and in vivo “ by Moustafa et al. describes the effects of HCV C from subtype 4a and 4f on TNF-induced cell death by using two stable cell lines and a transgenic mouse model. Previous studies on this topic have given conflicting results and the present study, which describes distinct effects that are subtype- and cell dependent, contributes to a better understanding and is therefore worthwhile publishing. The interpretation of data obtained from stable cell lines is complicated by the risk for random events due to clonal selection but although it would have been preferred to have data from sister clones, the coherent data from the transgenic mice partially alleviate this request. Data from gene silencing or gene induction that would bring further insights into a more mechanistic analysis of the observations described are not included.
Specific points.
1. Analysis of differential gene expression in the transgenic mice livers (Table 1-2 and Fig 3) is compelling but a corresponding analysis of the cell lines would strengthen the paper significantly.
2. An alignment of the HCV 4a and 4f core protein highlighting important differences would be helpful, in particular since the effects on gene expression is quite different (this is listed as Supplementary Figure but not found in the uploaded file). Possible explanations for this difference could preferably be added to the discussion.
Author Response
The manuscript entitled “ Hepatitis C core protein induces a genotype-specific susceptibility of hepatocytes to TNF-induced death in vitro and in vivo “ by Moustafa et al. describes the effects of HCV C from subtype 4a and 4f on TNF-induced cell death by using two stable cell lines and a transgenic mouse model. Previous studies on this topic have given conflicting results and the present study, which describes distinct effects that are subtype- and cell dependent, contributes to a better understanding and is therefore worthwhile publishing. The interpretation of data obtained from stable cell lines is complicated by the risk for random events due to clonal selection but although it would have been preferred to have data from sister clones, the coherent data from the transgenic mice partially alleviate this request. Data from gene silencing or gene induction that would bring further insights into a more mechanistic analysis of the observations described are not included.
We would like to thank the reviewer for taking the time to review our manuscript. The strength of our manuscript is indeed the assessment of TNF sensitivity in vivo. It is the first time this has been done outside the cell culture systems and also it is the first time two core proteins are compared side by side, giving more in vivo insights into the discrepancy that lasted 2 decades in the bibliography. We have scheduled crossings with knockout mice in order to assess the pathways in vivo.
Specific points.
- Analysis of differential gene expression in the transgenic mice livers (Table 1-2 and Fig 3) is compelling but a corresponding analysis of the cell lines would strengthen the paper significantly.
The cell lines were analysed using RNA seq due to the high cost of the experiments. We rather focused on the more physiologically significant liver biopsies that include all the signalling related to liver homeostasis and pathology. We have performed more experiments as other reviewers suggested on other mRNAs of the liver adding the component of cytokines, showing a pro-inflammatory profile possibly related to the SAA1 upregulation. We have further discussed this in the results and discussion sections comparing the outcome with previous studies in line with our findings.
- An alignment of the HCV 4a and 4f core protein highlighting important differences would be helpful, in particular since the effects on gene expression is quite different (this is listed as Supplementary Figure but not found in the uploaded file). Possible explanations for this difference could preferably be added to the discussion.
We have added the alignment in the supplementary file and discussed the role of the amino acid differences in the discussion section.
Round 2
Reviewer 2 Report
The authors responded to reviwers' queries.